# Unexpected Clinically Relevant Findings Detected via Computed Tomography in Patients with Severe Aortic Stenosis Who Are Candidates for Transcatheter Aortic Valve Replacement

**DOI:** 10.3390/jcm14020467

**Published:** 2025-01-13

**Authors:** Nicola Bianchi, Federica Frascaro, Luca Zanarelli, Federico Marchini, Federico Sanguettoli, Sofia Meossi, Matteo Serenelli, Alessandro Leone, Carlo Penzo, Carlo Tumscitz, Gianluca Campo, Rita Pavasini

**Affiliations:** 1Cardiology Unit, Azienda Ospedaliero Universitaria di Ferrara, 44124 Ferrara, Italyfederica.frascaro92@gmail.com (F.F.); luca.zanarelli@edu.unife.it (L.Z.); mrcfrc2@unife.it (F.M.); f.sanguettoli@gmail.com (F.S.); sofia.meossi@unife.it (S.M.); matteoserenelli@gmail.com (M.S.); carlopenzo78@gmail.com (C.P.); tumscitz@gmail.com (C.T.); cmpglc@unife.it (G.C.); 2Cardiology Unit, Ospedale “Degli Infermi” di Faenza, 48018 Faenza, Italy; 3Division of Cardiac Surgery, Cardiac Surgery Department, IRCCS Azienda Ospedaliero-Universitaria Di Bologna, S. Orsola Hospital, University of Bologna, 40138 Bologna, Italy; leone.alessandro@yahoo.it

**Keywords:** TAVI, TAVR, transcatheter aortic valve replacement, transcatheter aortic valve implantation, incidentaloma

## Abstract

**Background:** The detection of unexpected findings (UF) during CT scans of patients undergoing TAVR is frequent; however, it is unclear whether such findings have a clinical impact on the TAVR pathway. **Methods:** We conducted a retrospective, single-center observational study enrolling patients who were candidates for TAVR. All enrolled patients underwent a CT scan before valve implantation. The primary outcome of this study was all-cause mortality, while the secondary outcome was to determine whether the diagnosis of clinically relevant UF on CT scans results in a significant delay in the TAVR procedure. **Results:** A total of 284 patients were enrolled. Clinically relevant UF were identified in 15% of the patients, with the most common types being pulmonary masses or nodules. During the follow-up period, 83 patients (29.2%) died. The prognosis was worsened by chronic kidney disease (HR 1.76, *p* = 0.03) and left ventricular dilatation (HR 1.74, *p* = 0.04), while the diagnosis of clinically relevant UF did not impact all-cause mortality (*p* = 0.38). No statistically significant differences were found in the delay from the diagnosis of severe aortic stenosis to TAVR between patients with and without clinically relevant UF (*p* = 0.07), although patients with clinically relevant UF experienced a median delay of approximately 37 days in the TAVR procedure. **Conclusions:** The presence of clinically relevant UF on preoperative CT scans does not affect all-cause mortality but shows a trend toward increasing the time from diagnosis to the procedure in patients with severe aortic stenosis undergoing TAVR. Further studies are required to confirm these findings in larger patient cohorts.

## 1. Introduction

Aortic stenosis is the most commonly diagnosed valvular heart disease in Europe and North America, with its prevalence rapidly increasing due to the aging population [1,2]. Degenerative aortic stenosis is a complex condition driven by inflammation, lipid accumulation, and progressive calcification [3]. The prevalence of severe aortic stenosis in individuals over 75 years old is estimated to be approximately 3.5% [3]. Patients with symptomatic severe aortic stenosis face a high risk of cardiovascular mortality; if left untreated, the annual mortality rate is 25%, with an average survival of only 2 to 3 years [4,5].

According to the latest European Society of Cardiology guidelines [1], transcatheter aortic valve replacement (TAVR) is the treatment of choice for patients aged over 75 years or those at high risk for surgical replacement (STS-PROM/EuroSCORE II > 8%) when femoral TAVR is feasible.

TAVR requires meticulous pre-procedural planning. A comprehensive evaluation of the valve, perivalvular area, and vascular access is essential, typically performed with thoraco-abdominal computed tomography (CT) angiography [1,6]. Key parameters to assess include the size of the valvular annulus, the left ventricular outflow tract, the height of the coronary artery ostia, the sino-tubular junction, the ascending aorta, and peripheral vascular access [6].

However, preoperative CT scans in elderly patients undergoing TAVR may reveal unexpected findings, also known as incidentalomas. An incidentaloma is defined as an unexpected finding detected during imaging performed for another clinical reason [7,8,9]. Their prevalence and prognostic significance vary widely depending on the diagnostic examination and the organ involved [8,9]. These findings may range from benign lesions (e.g., cysts, lipomas, hernias) to nodules or masses of uncertain clinical significance. Such findings may necessitate additional diagnostic investigations, such as further imaging or specialist evaluations [8,9].

The management of these lesions in the context of pre-TAVR CT scans is critically important. If such findings are shown to influence patient prognosis, they could potentially alter or delay the TAVR pathway.

This study aims to describe the prevalence and types of unexpected findings (incidentalomas) detected on CT scans performed for TAVR. Additionally, it seeks to evaluate whether these findings serve as independent predictors of all-cause mortality and whether their diagnosis impacts the time to TAVR.

## 2. Materials and Methods

### 2.1. Healthcare Organization

The University Hospital of Ferrara is part of the public health organization of Emilia-Romagna, a region in Northern Italy. This hospital serves as one of the spoke centers for TAVR, where patients with severe aortic stenosis are screened, discussed by a multidisciplinary Heart Team, and then referred to a hub center with on-site cardiac surgery facilities for TAVR procedures.

Since January 2016, a dedicated care pathway has been established for patients diagnosed with severe aortic valve stenosis who are eligible for TAVR. Per hospital protocol and in accordance with guidelines [1], all patients considered for TAVR undergo comprehensive diagnostic evaluations, including echocardiography, coronary angiography, thoraco-abdominal CT scans, and laboratory analyses, before the Heart Team discussion. Following TAVR implantation, all patients are scheduled for follow-up visits.

The *Observational Study Relating to the Long-term Follow-up of Patients with Severe Aortic Stenosis Evaluated for Percutaneous Aortic Valve Replacement* (TAVI) (ROARING-TAVI) is a non-profit, observational study that includes all consecutive patients enrolled in the TAVR pathway since January 2016. This study does not alter or modify routine clinical practice and does not mandate the use of specific devices by any center or operator. It involves collecting pre- and post-procedural data from patients undergoing the TAVR pathway.

This study complies with the ethical principles outlined in the Declaration of Helsinki and received approval from the Ethical Review Board “Comitato Etico di Area Vasta Emilia Centro” on 12 September 2024, with ID: 519/2024/Oss/AOUFe. All patients provided informed written consent. This study is registered on ClinicalTrials.gov with ID: NCT06679517.

### 2.2. Study Population

This study analyzes retrospective data focusing on patients who (i) had a diagnosis of symptomatic severe aortic stenosis with an indication for aortic valve replacement; (ii) underwent all pre-procedural TAVR examinations (including CT scans); and (iii) had an indication for TAVR based on current guidelines. Exclusion criteria included (i) the inability to obtain complete retrospective data and (ii) exclusion from the TAVR pathway before completing pre-procedural examinations.

The diagnosis of severe aortic stenosis was established through echocardiographic assessment [10] and, when necessary, confirmed with additional methods (e.g., low-dose dobutamine stress echocardiography or aortic valve calcium score assessment) in accordance with current guidelines [10].

Medical treatment for participants was managed at the discretion of attending physicians, adhering to institutional protocols and international guidelines.

### 2.3. Clinically Relevant Unexpected Findings Definition

Unexpected findings (incidentalomas) were defined as any incidentally discovered mass or lesion detected on pre-TAVR CT scans. For the purposes of this study, these findings were categorized into two groups: (i) clinically relevant unexpected findings and (ii) benign unexpected findings that did not require additional examination.

Clinically relevant unexpected findings were defined as any previously unknown mass or lesion of uncertain clinical significance that required further anatomical or functional imaging or specialist medical evaluation. The goal of further examinations was to determine the nature of the lesion (benign or malignant) and assess whether the diagnosis of the unexpected finding could impact the patient’s prognosis, potentially altering their eligibility for the TAVR pathway.

### 2.4. Study Endpoints

The main objective of this study was to assess whether the identification of lesions with unclear clinical significance on pre-TAVR CT scans—requiring further diagnostic investigations (clinically relevant unexpected findings) before confirming or ruling out the feasibility of TAVR—had a significant impact on two critical aspects of the TAVR pathway: all-cause mortality and the prolongation of waiting time before TAVR.

The primary endpoint of this study was to identify independent predictors of all-cause mortality in a population of patients with severe aortic stenosis undergoing TAVR. In particular, this study aimed to verify whether the presence of clinically relevant unexpected findings on pre-TAVR CT scans was an independent predictor of all-cause mortality. Notably, patients with benign unexpected findings that did not require additional investigations were analyzed together with patients without unexpected findings, as their diagnoses did not necessitate further medical intervention.

The secondary endpoint was to determine whether the diagnosis of clinically relevant unexpected findings resulted in a significant delay in the TAVR procedure. The time to TAVR was calculated from the date of the last instrumental examination performed prior to the Heart Team discussion.

### 2.5. Data Collection

The following data were collected: demographic and anthropometric data, cardiovascular risk factors, major comorbidities, echocardiographic data, including a complete assessment of aortic stenosis, coronary artery angiography data, including coronary artery disease, any percutaneous coronary intervention (PCI) or percutaneous balloon aortic valvuloplasty performed, CT scan data, including the presence of clinically relevant unexpected findings, descriptions of these, and, if required, other clinical or imaging investigations following these findings, and the date of the TAVR procedure.

### 2.6. Statistical Analysis

Continuous variables were expressed as mean ± standard deviation (SD) if they were normally distributed or as median and interquartile range (IQR) if they were not. Normal distribution was assessed using the Shapiro–Wilk test. Categorical variables were expressed as counts and percentages.

Comparisons of normally distributed continuous variables were performed using Student’s *t*-test, while non-normally distributed continuous variables were compared using the Mann–Whitney U test. Dichotomous variables were compared using Fisher’s exact test.

Univariate and multivariate Cox regression analyses were conducted to evaluate all-cause mortality, including all variables listed in Table 1. Only variables with a *p*-value < 0.05 in the univariate analysis were entered into the multivariable model. Statistical analyses were performed using STATA version 15.0 (StataCorp LLC, College Station, TX, USA).

## 3. Results

### 3.1. Population Characteristics

A total of 284 patients were enrolled in this study. After a median follow-up period of 714 days, 83 patients (29.2%) had died. The mean age of the population was 87 years (range 83–90), and 151 patients (53%) were female. The cohort was stratified based on survival status at follow-up. Surviving patients (201, 70.1% of the population) were slightly but statistically significantly younger than non-surviving patients (86 years vs. 88 years, respectively; *p* = 0.036).

The most prevalent cardiovascular risk factors were arterial hypertension and dyslipidemia, which were present in 251 (88%) and 137 (49%) patients, respectively. There were no significant differences between the groups for these or any other cardiovascular risk factors (Table 1).

Among the major non-cardiovascular comorbidities analyzed (chronic kidney disease, oncologic disease, chronic obstructive pulmonary disease), only chronic kidney disease (CKD) was more frequent in the non-surviving group (27% vs. 51%, *p* < 0.01). Similarly, among cardiovascular comorbidities (atrial fibrillation, peripheral artery disease, and coronary artery disease), atrial fibrillation (AF) was significantly more common in the non-surviving group (38% vs. 56%, *p* = 0.01) (Table 1).

There were no significant differences between the groups in major cardiovascular procedures performed during the index coronary angiography before TAVR, such as percutaneous coronary intervention or balloon aortic valvuloplasty (Table 1).

### 3.2. Echocardiographic Characteristics

A full transthoracic echocardiogram, including a complete assessment of aortic stenosis, was performed on all patients. In non-surviving patients, the mean systolic blood pressure was slightly but statistically significantly lower (43 mmHg vs. 42 mmHg, *p* = 0.03) compared to surviving patients. Additionally, left ventricular ejection fraction (EF) was significantly lower in non-surviving patients (58% vs. 55%, *p* = 0.02), and left ventricular dilation was more frequent in this group (17% vs. 31%, *p* = 0.02).

### 3.3. Unexpected Findings at CT Scan

Unexpected findings on the CT scan were detected in 57% of patients, with clinically relevant unexpected findings observed in 43 patients (15%). There was no significant difference between surviving and non-surviving patients, either for those with clinically relevant unexpected findings or for those with benign unexpected findings (*p* = 0.70 and *p* = 0.77, respectively).

The types of benign unexpected findings that did not require additional examination are described in Table 2.

The most common clinically relevant unexpected findings requiring additional specialist evaluations were pulmonary nodules or masses (32% of clinically relevant unexpected findings), followed by renal masses and pancreatic masses (14% each) and adrenal masses (12%). The remaining 28% of clinically relevant unexpected findings included suspected hematological diseases, gastrointestinal masses, thyroid masses, ovarian masses, and urinary bladder masses, as shown in Figure 1. This figure also details the further investigations required following the diagnosis.

No differences were found between patients with clinically relevant unexpected findings on the CT scan and patients without clinically relevant unexpected findings (including those with completely negative CT scans for unexpected findings) for any of the clinical characteristics (Table 3).

After stratification according to the presence or absence of incidentalomas, no differences were found in echocardiographic parameters between the groups.

### 3.4. Follow-Up and TAVR

A total of 284 patients were screened for eligibility for TAVR; of these, 235 (82%) were eligible and 49 (18%) were not. Among the eligible patients, 200 (85%) did not have any clinically relevant unexpected findings on the CT scan, and 30 (15%) of them died during follow-up. On the other hand, 35 (15%) had clinically relevant unexpected findings on the CT scan, and 7 (20%) died during follow-up.

Among the non-eligible patients, 41 (84%) did not have clinically relevant unexpected findings on the CT scan, and 39 (95%) of them died during follow-up. In contrast, 8 (16%) had clinically relevant unexpected findings on the CT scan, and 7 (88%) died during follow-up (graphical abstract).

TAVR was performed in 165 patients (58%), with a significant difference between non-surviving and surviving patients (45% vs. 64%, respectively; *p* < 0.01). The median time from diagnosis with all necessary exams for Heart Team discussion to the TAVR procedure was 169 [131–234] days, with no statistically significant difference between non-surviving and surviving patients.

### 3.5. Primary Endpoint Analysis

As a result of the primary endpoint analysis, no statistically significant effect of clinically relevant unexpected findings on the CT scan on all-cause mortality was found (*p*-value = 0.38, HR = 1.33, 95% CI 0.71–2.48; Figure 2). However, in the multivariable model (Table 4), chronic kidney disease (CKD) was significantly associated with a worse prognosis (HR 1.76; *p* = 0.03), as was left ventricular (LV) dilatation (HR 1.74; *p* = 0.04). Conversely, undergoing TAVR was associated with a reduction in the risk of all-cause mortality (HR = 0.35; *p* < 0.01).

### 3.6. Secondary Endpoint Analysis

The median delay from diagnosis to TAVR was 153 days [IQR 91–214 days] in patients without clinically relevant unexpected findings on the CT scan and 190 days [IQR 142–279 days] in patients with clinically relevant unexpected findings on the CT scan (Figure 3). There was no statistically significant difference between the groups (*p* = 0.07). Although the result was not statistically significant, a trend toward an increased median delay of 37 days was observed.

## 4. Discussion

This retrospective, observational, monocentric study on patients with severe aortic stenosis indicated for TAVR showed the following:(i) The incidence of clinically relevant unexpected findings found on preoperative CT scans for TAVR is 15% (43/284).(ii) The most common clinically relevant unexpected finding is the presence of one or more pulmonary nodules.(iii) Identifying clinically relevant unexpected findings does not affect all-cause mortality in patients with severe aortic stenosis who are candidates for TAVR, but it is associated with a trend toward increased delay from diagnosis to procedure.

The improvement in well-being and the consequent increase in life expectancy have led to a steady rise in the number of patients affected by severe aortic stenosis who are awaiting TAVR [2]. This issue has been further exacerbated by the fact that even patients with intermediate and low risk for severe aortic stenosis can now be treated with TAVR [1].

Previous studies have shown that long waiting times for TAVR are associated with increased mortality, all-cause hospitalization, and worsening of functional status and quality of life [11,12,13]. Waiting longer than 60 days can result in a loss of the benefit of TAVR compared to surgery [14]. Recently, it has been demonstrated that providing faster access to TAVR for higher-risk patients results in a 29% reduction in mortality, a 23% reduction in hospitalizations, and a 38% reduction in urgent procedures [15,16]. However, to achieve these results, wait times should be within 3 weeks for high-risk patients and 7 weeks for medium-risk patients [15,16]. In practice, however, these estimates are not easily achievable. Waiting times for patients with severe aortic stenosis vary widely across different healthcare systems, even within the same country, and even more so when comparing different countries [4,5,11]. In Canada, the mean waiting time for elective TAVR is 148.5 ± 118.5 days [11], while in Algeria, for example, it is around 231.7 ± 134.1 days [16]. In our study, the mean waiting time was 169 [131–234] days.

These observations reflect a significant and persistent issue related to TAVR procedures in Italian spoke centers. The rising incidence and prevalence of aortic stenosis in the general population have also significantly increased waiting lists and exacerbated inequities in access to therapies [2], especially for patients referred from various spoke centers to a single hub center, leading to waiting times of over 5–6 months for each patient for elective TAVR. Our study focuses on an important issue that could be a contributing factor to the increase in TAVR waiting times: the discovery of clinically relevant unexpected findings on CT scans. However, our data showed that clinically relevant unexpected findings are relatively common in the elderly population undergoing TAVR, but their presence does not significantly influence the waiting time for the procedure, resulting in a non-significant increase of 37 days compared to patients without incidentalomas.

These results, along with data from the multivariate analysis showing TAVR as an important prognostic factor, underscore how aortic stenosis is a crucial determinant of prognosis in elderly patients, even in comparison to other potential comorbidities (e.g., newly identified potentially malignant masses). Among these, CKD and left ventricular dilatation were independently associated with poor prognosis in the multivariate analysis. Aortic stenosis progresses more rapidly and is more prevalent in patients with CKD, who also experience worse outcomes and higher mortality [17,18,19,20]. Finally, left ventricular dilatation represents the most severe stage in the progression of heart failure in patients with severe aortic stenosis [21,22,23], as confirmed by our study data, which reinforce its negative prognostic impact.

Joseph et al. [5] and Pellikka et al. [24] have previously shown that treating aortic stenosis with valve replacement significantly improves life expectancy in patients with severe aortic stenosis, particularly once the patient becomes symptomatic, as the prognosis worsens. Data from this study confirm and expand these findings, showing that clinically relevant unexpected findings detected on pre-TAVR CT scans do not have a worse prognostic impact than prolonged waiting times for TAVR.

All these findings prompt reflection on possible organizational solutions to reduce TAVR waiting times, such as conducting randomized clinical trials to evaluate the feasibility of performing TAVR in spoke centers with experienced TAVR operators but without backup cardiac surgery [25,26,27].

### Study Limitations

The main limitations of this study in assessing the prognostic impact of clinically relevant unexpected findings on CT scans include the long waiting period between the diagnosis of severe aortic stenosis and the TAVR procedure, as well as the small sample size. Additionally, a specific evaluation of patient frailty was not available. Further prospective multicenter studies are needed to confirm our findings.

## 5. Conclusions

In conclusion, the presence of clinically relevant unexpected findings on preoperative CT scans is not an independent predictor of all-cause mortality in patients with aortic stenosis undergoing TAVR. However, it is associated with a trend toward longer waiting times from diagnosis to procedure.

## Figures and Tables

**Figure 1 jcm-14-00467-f001:**
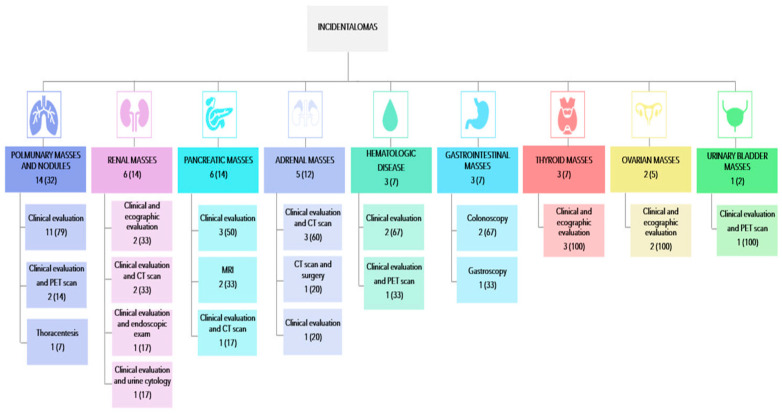
Types and frequencies of CT incidentalomas found and types of further investigation needed after their diagnosis. CT: computed tomography; MRI: magnetic resonance; PET: positron emission tomography. Percentages are indicated in round brackets.

**Figure 2 jcm-14-00467-f002:**
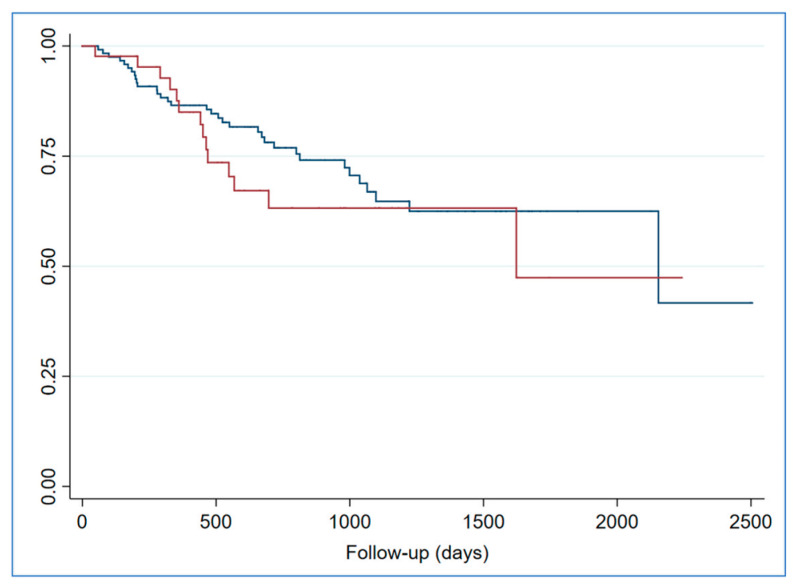
Kaplan–Meier survival estimates in patients with (red line) or without (blue line) clinically relevant unexpected findings on CT scan. Wilcoxon–Breslow–Gehan test *p*-value = 0.37.

**Figure 3 jcm-14-00467-f003:**
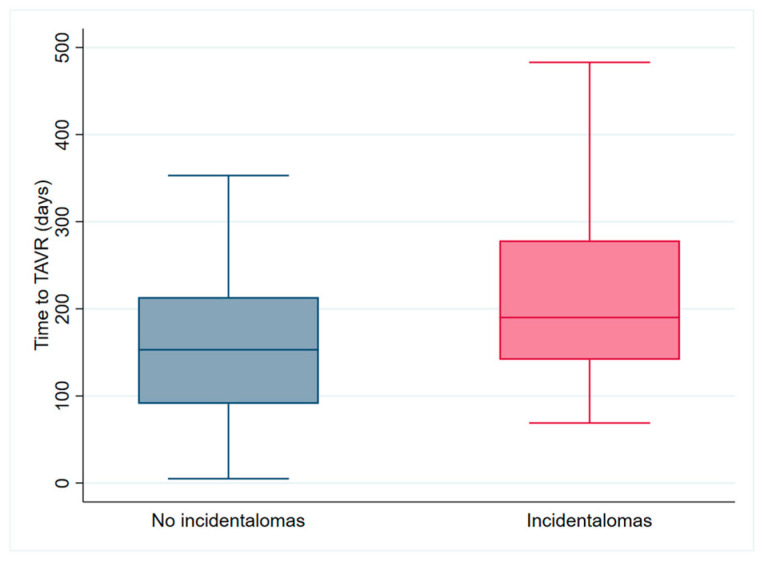
Bar chart of time to TAVR procedure in patients with (red bar) or without (blue bar) clinically relevant unexpected finding on CT scan. Mann–Whitney U test *p*-value = 0.07.

**Table 1 jcm-14-00467-t001:** Study population characteristics by patient survival.

	Overall	Survived	Did Not Survive	*p*-Value
	N = 284	N = 201	N = 83	
Age (years)	87 [83–90]	86 [83–90]	88 [84–90]	**0.04**
BSA (kg/m^2^)	1.78 ± 0.21	1.77 ± 0.21	1.80 ± 0.19	0.32
Female sex	151 (53)	114 (57)	37 (45)	0.07
**Comorbidity and procedures**				
CKD	95 (34)	53 (27)	42 (51)	**<0.01**
COPD	39 (14)	28 (14)	11 (13)	1.00
Neoplasm	63 (22)	41 (21)	22 (27)	0.28
PAD	30 (10)	19 (10)	11 (13)	0.40
AF	123 (44)	77 (38)	46 (56)	**0.01**
CAD	187 (66)	130 (65)	57 (69)	0.68
PCI	133 (47)	93 (47)	40 (48)	0.90
Ballon aortic valvuloplasty	195 (69)	135 (67)	60 (73)	0.48
**Cardiovascular risk factors**				
Family history of CAD	14 (5)	12 (6)	2 (2)	0.36
Diabetes	82 (29)	56 (28)	26 (32)	0.57
Dyslipidemia	137 (49)	99 (50)	38 (46)	0.60
Smoke (current or former)	93 (33)	62 (31)	31 (38)	0.33
Arterial hypertension	251 (88)	179 (89)	72 (86)	0.52
**Echocardiography**				
Maximum systolic TG (mmHg)	69 ± 19	69 ± 18	67 ± 19	0.28
Mean systolic TG (mmHg)	43 [35–49]	43 [36–50]	42 [32–46]	**0.03**
Effective orifice area (cm^2^)	0.8 [0.7–0.9]	0.8 [0.7–0.9]	0.8 [0.7–0.9]	0.21
LV dilatation	61 (21)	35 (17)	26 (31)	**0.02**
EF (%)	55 [45–62]	58 [50–64]	55 [41–60]	**0.02**
Severe MR	17 (6)	8 (4)	9 (11)	0.06
Severe MS	2 (1)	1 (1)	1 (1)	0.50
Severe AR	18 (6)	13 (7)	5 (6)	1.00
**Unexpected findings on CT scan**				
Unexpected findings	163 (57)	115 (57)	48 (58)	1.00
Clinically relevant	43 (15)	29 (14)	14 (17)	0.70
Benign unexpected findings	120 (42)	86 (43)	34 (41)	0.77
**TAVR and follow-up**				
TAVR procedure performed	165 (58)	128 (64)	37 (45)	**<0.01**
Exclusion from TAVR	49 (18)	3 (2)	46 (55)	**<0.01**
Study follow-up (days)	714 [464–1131]	821 [530–1258]	486 [255–938]	**<0.01**
TAVR procedure delay (days)	169 [131–234]	143 [102–206]	169 [112–245]	0.19

BSA: body surface area; CKD: chronic kidney disease; COPD: chronic obstructive pulmonary disease; PAD: peripheral artery disease; AF: atrial fibrillation; CAD: coronary artery disease; PCI: percutaneous coronary intervention; TG: transvalvular gradient; LV: left ventricle; EF: ejection fraction; MR: mitral regurgitation; MS: mitral stenosis; AR: aortic regurgitation; TAVR: transcatheter aortic valve replacement; []: interquartile range; (): percentage; *p* < 0.05 are in bold.

**Table 2 jcm-14-00467-t002:** Benign unexpected findings on the CT scan pre-TAVR that did not require additional examination.

Benign Unexpected Finding	N (%)
Baker’s cyst	2 (1.7)
Diverticula of the sigmoid colon	31 (25.8)
Osteoarthritis	11 (9.2)
Gallstones	12 (10)
Lipoma	1 (0.8)
Benign prostatic hyperplasia	14 (11.7)
Detrusor muscle hypertrophy	4 (3.4)
Hepatic hemangioma	8 (6.8)
Uncomplicated inguinal hernia	3 (2.6)
Spondylolisthesis	1 (0.8)
Bladder diverticulum	3 (2.6)
Simple renal cysts	2 (1.7)
Bovine aortic arch	3 (2.6)
Abdominal rectus diastasis	1 (0.8)
Uncomplicated laparocele	1 (0.8)
Reactive lymph nodes	3 (2.6)
Varicocele	1 (0.8)
Uterine fibroid	1 (0.8)
Horseshoe kidney	1 (0.8)
Hiatal hernia	10 (8)
Right renal artery duplication	1 (0.8)
Hip osteoarthritis	5 (4)
Zenker’s diverticulum	1 (0.8)

N: number of lesions; %: percentage.

**Table 3 jcm-14-00467-t003:** Study population characteristics: patients with or without clinically relevant incidentaloma.

	Without Clinically Relevant Unexpected Findings on CT Scan	With Clinically Relevant Unexpected Findings on CT Scan	*p*-Value
	N = 241	N = 43	
Age (years)	87 [83–90]	86 [83–90]	0.96
BSA (kg/m^2^)	1.79 ± 0.21	1.75 ± 0.23	0.31
Female sex	125 (52)	26 (60)	0.11
Comorbidity and procedures			
CKD	79 (33)	16 (38)	0.60
COPD	33 (14)	6 (14)	1.00
Neoplasm	52 (22)	11 (26)	0.55
PAD	26 (11)	4 (9)	1.00
AF	108 (46)	15 (35)	0.24
CAD	115 (48)	24 (56)	1.00
PCI	115 (48)	18 (42)	0.51
Ballon aortic valvuloplasty	161 (68)	34 (81)	0.10
Cardiovascular risk factors			
Family history of CAD	12 (5)	2 (5)	1.00
Diabetes	68 (29)	14 (33)	0.59
Dyslipidemia	118 (49)	19 (44)	0.51
Smoke	80 (34)	13 (30)	0.73
Arterial hypertension	211 (89)	40 (93)	0.59
Echocardiography			
Maximum systolic TG (mmHg)	69 ± 18	68 ± 21	0.82
Mean systolic TG (mmHg)	42 [35–49]	43 [37–50]	0.85
Effective orifice area (cm^2^)	0.8 [0.7–0.9]	0.8 [0.7–0.9]	0.96
LV dilatation	55 (23)	6 (14)	0.30
EF (%)	56 [46–62]	55 [45–64]	0.88
Severe MR	14 (6)	3 (7)	0.73
Severe MS	2 (1)	0 (0)	1.00
Severe AR	14 (6)	4 (9)	0.49
TAVR and follow-up			
TAVR procedure performed	143 (59)	22 (51)	0.32
Exclusion from TAVR	41 (18)	8 (19)	0.83
Study follow-up (days)	737 [481–1186]	568 [373–984]	0.06
TAVR procedure delay (days)	153 [91–214]	190 [142–279]	0.07

BSA: body surface area; CKD: chronic kidney disease; COPD: chronic obstructive pulmonary disease; PAD: peripheral artery disease; AF: atrial fibrillation; CAD: coronary artery disease; PCI: percutaneous coronary intervention; TG: transvalvular gradient; LV: left ventricle; EF: ejection fraction; MR: mitral regurgitation; MS: mitral stenosis; AR: aortic regurgitation; TAVR: transcatheter aortic valve replacement. []: interquartile range; (): percentage.

**Table 4 jcm-14-00467-t004:** Univariate and multivariate Cox regression models.

**Univariate Analysis**			
	**HR**	**CI 95%**	***p*-Value**
Age (years)	1.01	0.97–1.05	0.71
CKD	2.02	1.33–3.17	**<0.01**
AF	1.67	1.08–2.59	**0.02**
Mean systolic TG (mmHg)	0.97	0.96–0.99	**0.03**
LV dilatation	1.93	1.20–3.10	**<0.01**
EF (%)	0.98	0.97–1.01	0.06
TAVR procedure performed	0.46	0.30–0.71	**<0.01**
Exclusion from TAVR	9.62	6.21–14.9	**<0.01**
Clinically relevant unexpected finding on CT scan	1.33	0.71–2.48	0.38
**Multivariate Analysis**			
CKD	1.76	1.08–2.89	**0.03**
AF	1.54	0.94–2.54	0.08
Mean systolic TG	0.98	0.96–1.01	0.10
LV dilatation	1.74	1.01–3.03	**0.04**
TAVR procedure	0.35	0.22–0.57	**<0.01**

CKD: chronic kidney disease; AF: atrial fibrillation; TG: transvalvular gradient; LV: left ventricle; TAVR: transcatheter aortic valve replacement; EF: ejection fraction; *p* < 0.05 are in bold.

## Data Availability

The data presented in this study are available on reasonable request from the corresponding author.

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
