# Peer review of "Unexpected Clinically Relevant Findings Detected via Computed Tomography in Patients with Severe Aortic Stenosis Who Are Candidates for Transcatheter Aortic Valve Replacement"

_jcm, 2025, doi:10.3390/jcm14020467_

Round 1

Reviewer 1 Report

Comments and Suggestions for Authors

The authors of the manuscript present results from a retrospective, single-center observational study that enrolled patients with severe aortic stenosis who were indicated for TAVR and had a pre-procedural CT scan. The study aimed to explore the clinical significance of incidental findings, or incidentalomas, identified in these CT scans. The manuscript is relevant, as degenerative aortic stenosis is common among the elderly, and its incidence is increasing in many countries due to population aging. Severe degenerative aortic stenosis is typically the leading indication for valve replacement in most countries. On the other hand, incidentalomas are also common, and the clinical significance of these incidental findings is often unclear in many patients.

The overall assessment of the manuscript is as follows: The introduction does not provide sufficient background.  The research design is not appropriate to the study real objectives (evaluation of clinical significance of incidentalomas in patients with severe aortic stenosis referred for TAVR). The methods are described plain. The results are clearly presented but most of them are not relevant to the study real objectives (clinical significance of incidentalomas). The conclusions are not supported by the results. The English language must be improved. 

I have the following comments, questions and recommendations regarding the manuscript:

  1. The title of the manuscript ("Prognostic role of extracardiac incidentalomas...") does not closely align with the primary and secondary outcomes of the study as stated in the abstract ("The primary outcome of the study was all-cause mortality, and the secondary outcome was the time to the TAVR procedure"). These outcomes are likely influenced by factors other than extracardiac incidentalomas in this study population. Therefore, the title and outcomes (or objectives, or primary/secondary endpoints) should be revised to better reflect the study's focus.

  2. The background of the abstract does not mention incidentalomas, focusing only on aortic stenosis and TAVR. However, the title implies that the manuscript will primarily address the clinical significance of extracardiac incidentalomas. This discrepancy should be addressed for consistency.

  3. The term "pulmonary masses or nodules" identified in this study does not necessarily equate to incidentalomas. An incidentaloma refers to a radiological finding of a lesion discovered incidentally, which may have dubious clinical significance. It is commonly used for lesions found in endocrine glands (e.g., adrenal glands) and is most often associated with adrenal adenomas. Although the term is sometimes used more broadly, referring to any incidental lesion (e.g., pituitary, thyroid), its use in relation to pulmonary masses or nodules could be misleading and should be avoided.

  4. What do the authors mean by "clinically relevant incidentalomas"? Since incidentalomas are typically asymptomatic, what criteria did the authors use to determine the clinical significance of these findings?

Comments on the Quality of English Language

The English language must be improved. 

Author Response

Reviewer #1

Comment#1

The authors of the manuscript present results from a retrospective, single-center observational study that enrolled patients with severe aortic stenosis who were indicated for TAVR and had a pre-procedural CT scan. The study aimed to explore the clinical significance of incidental findings, or incidentalomas, identified in these CT scans. The manuscript is relevant, as degenerative aortic stenosis is common among the elderly, and its incidence is increasing in many countries due to population aging. Severe degenerative aortic stenosis is typically the leading indication for valve replacement in most countries. On the other hand, incidentalomas are also common, and the clinical significance of these incidental findings is often unclear in many patients.

Reply #1

First of all, we would like to thank the Reviewer #1 for having understand the importance of the main issue of the present study. We would also like to thank the Reviewer #1 for the precious comments that will help us improve the clarity of the exposition of the contents and make the objectives of our study more understandable. We have tried to respond to all the comments made. Thanks again.

Comment #2

The overall assessment of the manuscript is as follows: The introduction does not provide sufficient background.

Reply #2

We thank the reviewer for the comment. We have better explain in introduction what incidentalomas are and which were the objective of the study.

Modified text: Section Introduction, page 2, lines 53-67

However, preoperative CT scans in elderly patients undergoing TAVR may reveal unexpected findings, also known as incidentalomas. An incidentaloma is defined as an unexpected finding detected during imaging performed for another clinical reason [8,9]. Their prevalence and prognostic significance vary widely depending on the diagnostic examination and the organ involved [8,9]. These findings may range from benign lesions (e.g., cysts, lipomas, hernias) to nodules or masses of uncertain clinical significance. Such findings may necessitate additional diagnostic investigations, such as further imaging or specialist evaluations [8,9].

The management of these lesions in the context of pre-TAVR CT scans is critically important. If such findings are shown to influence patient prognosis, they could potentially alter or delay the TAVR pathway.

This study aims to describe the prevalence and types of unexpected findings (incidentalomas) detected on CT scans performed for TAVR. Additionally, it seeks to evaluate whether these findings serve as independent predictors of all-cause mortality and whether their diagnosis impacts the time to TAVR.

Comment#3

 The research design is not appropriate to the study real objectives (evaluation of clinical significance of incidentalomas in patients with severe aortic stenosis referred for TAVR). The methods are described plain. The results are clearly presented but most of them are not relevant to the study real objectives (clinical significance of incidentalomas). The conclusions are not supported by the results.

Reply #3

We are sorry for having been not precise and clear enough to making understanding to the reader the objectives of our study. Incidentalomas are commonly described as any incidental finding during a diagnostic examination (in this case the pre-TAVR CT scan) of anatomical alterations that may be benign, but that sometimes require evaluation by specialists in the field to understand their clinical significance. As described in the methods, we defined "clinically relevant incidentalomas" as all incidental findings detected at the pre-TAVR CT scan that needed a further clinical or imaging investigations. In particular all pre-TAVR CT scans are evaluated in a specific valve clinic, where the planning of the procedure is done using CT imaging. Once unexpected findings have been described into the CT examination, the Cardiology consultant who is in charge of the patient evaluated whether the incidentaloma could require further evaluation by the relevant specialist or further specific imaging depending on the lesion identified, before the TAVR to be planned, because the incidentalomas might have a prognostic impact on patient. To be more precise, in this version of the manuscript we have better described also incidentalomas not needing other specific analyses. We also implemented the definition of "clinically relevant incidentalomas" into the methods.

The main objective of the study was to understand whether the recognition of the presence of lesions of non-univocal clinical significance on pre-TAVR CT and therefore requiring further examinations or diagnostics before confirming or not the feasibility of TAVR, had a significant impact on two of the main nodes of the TAVR pathway: all-cause mortality and the lengthening of the waiting time before TAVR. These two factors are closely linked, considering that the literature data confirm that one of the factors that most affects the prognosis of patients with severe aortic stenosis is the waiting time before the procedure. Our objective was therefore to understand whether the presence of clinically significance unexpected finding at CT scan pre-TAVR was an independent predictor of all-cause mortality and if their diagnosis affectstime to TAVR. We have better explain the objective both in introduction and in methods, we also have better formulate the conclusions. Thank you for your advice.

Modified text: section Introduction, page 2, lines 64-67

This study aims to describe the prevalence and types of unexpected findings (incidentalomas) detected on CT scans performed for TAVR. Additionally, it seeks to evaluate whether these findings serve as independent predictors of all-cause mortality and whether their diagnosis impacts the time to TAVR.

Modified text: section Methods, page 3, lines 103-129, page 4, lines 130-133

Unexpected findings (incidentalomas) were defined as any incidentally discovered mass or lesion detected on pre-TAVR CT scans. For the purposes of this study, these findings were categorized into two groups: i) clinically relevant unexpected findings, and ii) benign unexpected findings that did not require additional examination.

Clinically relevant unexpected findings were defined as any previously unknown mass or lesion of uncertain clinical significance that required further anatomical or functional imaging or specialist medical evaluation. The goal of further examinations was to determine the nature of the lesion (benign or malignant) and assess whether the diagnosis of the unexpected finding could impact the patient’s prognosis, potentially altering their eligibility for the TAVR pathway.

2.4 Study endpoints

The main objective of the study was to assess whether the identification of lesions with unclear clinical significance on pre-TAVR CT scans—requiring further diagnostic investigations (clinically relevant unexpected findings) before confirming or ruling out the feasibility of TAVR—had a significant impact on two critical aspects of the TAVR pathway: all-cause mortality and the prolongation of waiting time before TAVR.

The primary endpoint of the study was to identify independent predictors of all-cause mortality in a population of patients with severe aortic stenosis undergoing TAVR. In particular, the study aimed to verify whether the presence of clinically relevant unexpected findings on pre-TAVR CT scans was an independent predictor of all-cause mortality. Notably, patients with benign unexpected findings that did not require additional investigations were analyzed together with patients without unexpected findings, as their diagnoses did not necessitate further medical intervention.

The secondary endpoint was to determine whether the diagnosis of clinically relevant unexpected findings resulted in a significant delay to the TAVR procedure. Time to TAVR was calculated from the date of the last instrumental examination performed prior to the Heart Team discussion.

Modified text: section Methods, page 4, lines 151-154

Univariate and multivariate Cox regression analyses were conducted to evaluate all-cause mortality, including all variables listed in Table 3. Only variables with a p-value < 0.05 in the univariate analysis were entered into the multivariable model.

Modified text: added Table 2

Modified text: section Conclusions, page 12, lines 332-335

In conclusion, the presence of clinically relevant unexpected findings on preoperative CT scans is not an independent predictor of all-cause mortality in patients with aortic stenosis undergoing TAVR. However, it is associated with a trend toward longer waiting times from diagnosis to procedure.

Comment #4

The English language must be improved. 

Reply #4

We are sorry if the English quality of the manuscript was not sufficient. We have sent out the new version of the manuscript to a native speaker for English revision.

Comment #5

I have the following comments, questions and recommendations regarding the manuscript:

  1. The title of the manuscript ("Prognostic role of extracardiac incidentalomas...") does not closely align with the primary and secondary outcomes of the study as stated in the abstract ("The primary outcome of the study was all-cause mortality, and the secondary outcome was the time to the TAVR procedure"). These outcomes are likely influenced by factors other than extracardiac incidentalomas in this study population. Therefore, the title and outcomes (or objectives, or primary/secondary endpoints) should be revised to better reflect the study's focus.

Reply #5

We thank the Reviewer for the comment and we regret for being not enough clear. As explained in Reply #3 the main objective of the study was to understand whether the recognition of the presence of lesions of non-univocal clinical significance on pre-TAVR CT and therefore requiring further examinations or diagnostics before confirming or not the feasibility of TAVR, had a significant impact on two of the main nodes of the TAVR pathway: all-cause mortality and the lengthening of the waiting time before TAVR. These two factors are closely linked, considering that the literature data confirm that one of the factors that most affects the prognosis of patients with severe aortic stenosis is the waiting time before the procedure. Our objective was therefore to understand whether the presence of clinically significance unexpected finding at CT scan pre-TAVR was an independent predictor of all-cause mortality and if their diagnosis affectstime to TAVR. We found that even if a patient in the TAVR pathway needs to do other medical examinations secondarily to the finding of unexpected lesions on the pre-TAVR CT scan, compared to those who are in the pathway but do not have incidentalomas (or those found do not require further analysis), they do not have a significant increase in mortality, probably because even if the average waiting times for TAVR tend to be longer, these are not significantly longer compared to those who do not need to do further investigations. We have done a specific uni- and multivariate analysis regarding the primary endpoint to understand wheatear the presence of “clinically relevant unexpected findings” might have been independent prognostic factors for influencing all-cause mortality. This was not the case: the significance is that the presence of clinically relevant incidentalomas at the CT-scan preTAVR are not factors that impact on mortality of patients that are already in the TAVR pathway compared to patients that do not need to do other medical investigations before the procedure. We have better underlined this concept also in abstract.

Following the suggestion of the Reviewer we have also changed the title.

Modified text: Title

Clinically relevant unexpected findings detected with computed tomography in patients with severe aortic stenosis candidates for transcatheter aortic valve replacement

Modified text: abstract

Methods:  We conducted a retrospective, single-center observational study enrolling patients who were candidates for TAVR. All enrolled patients underwent a CT scan before valve implantation. The primary outcome of the study was all-cause mortality, while the secondary outcome was to determine whether the diagnosis of clinically relevant UF on CT scans results in a significant delay in the TAVR procedure.

Comment #6

  1. The background of the abstract does not mention incidentalomas, focusing only on aortic stenosis and TAVR. However, the title implies that the manuscript will primarily address the clinical significance of extracardiac incidentalomas. This discrepancy should be addressed for consistency.

Reply #6

Reviewer #1 is right and we thank the Reviewer for the suggestion. We have changed the background of the abstract as suggested, focusing on incidentalomas.

Modified text: abstract

The detection of unexpected findings (UF) during CT scans of patients undergoing TAVR is frequent; however, it is unclear whether such findings have a clinical impact on the TAVR pathway.

Comment #7

  1. The term "pulmonary masses or nodules" identified in this study does not necessarily equate to incidentalomas. An incidentaloma refers to a radiological finding of a lesion discovered incidentally, which may have dubious clinical significance. It is commonly used for lesions found in endocrine glands (e.g., adrenal glands) and is most often associated with adrenal adenomas. Although the term is sometimes used more broadly, referring to any incidental lesion (e.g., pituitary, thyroid), its use in relation to pulmonary masses or nodules could be misleading and should be avoided.

Reply #7

We thank the Reviewer for this comment. According to the suggestion of the Reviewer we changed the term “incidentalomas” with “unexpected findings”. We hope this term might be more inclusive and therefore more appropriate to describe what in the previous version of the manuscript was defined “incidentalomas”.

Modified text

Word “incidentalomas” in all the manuscript replaced by “unexpected findings”.

Comment #8

  1. What do the authors mean by "clinically relevant incidentalomas"? Since incidentalomas are typically asymptomatic, what criteria did the authors use to determine the clinical significance of these findings?

Reply #8

As replied before, we regret for having been not enough clear about this point. As described in the methods, we defined "clinically relevant incidentalomas" as all incidental findings detected at the pre-TAVR CT scan that needed a further clinical or imaging investigations. In particular all pre-TAVR CT scans are evaluated in a specific valve clinic, where the planning of the procedure is done using CT imaging. Once unexpected findings have been described into the CT examination, the Cardiology consultant who is in charge of the patient evaluated whether the incidentaloma could require further evaluation by the relevant specialist or further specific imaging depending on the lesion identified, before the TAVR to be planned, because the incidentalomas might have a prognostic impact on patient. To be more precise, in this version of the manuscript we have attached a table describing the unexpected findings that were defined as clinically relevant from those that were instead judged not clinically relevant. We also implemented the definition of "clinically relevant unexpected findings" into the methods.

Modified text: section Methods, page 3, lines 106-116

Unexpected findings (incidentalomas) were defined as any incidentally discovered mass or lesion detected on pre-TAVR CT scans. For the purposes of this study, these findings were categorized into two groups: i) clinically relevant unexpected findings, and ii) benign unexpected findings that did not require additional examination.

Clinically relevant unexpected findings were defined as any previously unknown mass or lesion of uncertain clinical significance that required further anatomical or functional imaging or specialist medical evaluation. The goal of further examinations was to determine the nature of the lesion (benign or malignant) and assess whether the diagnosis of the unexpected finding could impact the patient’s prognosis, potentially altering their eligibility for the TAVR pathway.

Modified text: Table 2

Reviewer 2 Report

Comments and Suggestions for Authors

Congrats on this interesting idea and well-written manuscript. 

Some images of the findings would be an extra noteworthy addition to this article.

Author Response

Comment #1

Congrats on this interesting idea and well-written manuscript. 

Some images of the findings would be an extra noteworthy addition to this article.

Reply #1

We really thank the Reviewer for the positive comments. We have added a table describing what not clinically relevant unexpected findings are.

Modified text: Table 2

Reviewer 3 Report

Comments and Suggestions for Authors

Thank you for inviting me to review the paper by Nicola Bianchi et al. 

The project concerns an important issue in patients undergoing qualification for TAVI - incidentaloma in CT scans. 

The issue is important and worth describing.

The manuscript however needs some clarification. 

The authors describe supplementary data; however, I could not find one. 

It is not clear if patients with clinically signidficant incidentaloma had worse prognosis than those who had incidentaloma but not significant.

\Please correct the statement or sequence of presented age (Patients who survived (201 patients, 70.1% of the overall population) were 134 slightly but statistically significantly younger than those who did not survive (88 years 135 vs. 86 years, respectively; p=0.036). )

Please change comas into periods in tables 

I hope this points would help improving the paper 

Author Response

Comment #1

Thank you for inviting me to review the paper by Nicola Bianchi et al. 

The project concerns an important issue in patients undergoing qualification for TAVI - incidentaloma in CT scans. 

The issue is important and worth describing.

The manuscript however needs some clarification. 

Reply #1

We thank the Reviewer for having appreciated our manuscript and we thank the Reviewer for the precious comments with which we tried to improve the clarity and quality of the manuscript. Thank you.

Comment #2

The authors describe supplementary data; however, I could not find one. 

Reply #2

We are sorry for the mistake. The supplemental online Figure 1s is the graphical abstract. We changed the caption in the text.

Modified text: Page 7, lines 195

(Graphical abstract).

Comment #3

It is not clear if patients with clinically significant incidentaloma had worse prognosis than those who had incidentaloma but not significant.

Reply #3

We thank the Reviewer for this comment. Among patients who died at the follow-up, 14 had clinically relevant unexpected findings and 34 had benign unexpected findings, for both groups there was no difference between surviving and non-surviving patients. We have added a line in Table 1.

Modified text: Table 1

Modified text: Section Results, page 6, lines 191-194

There was no significant difference between surviving and non-surviving patients, either for those with clinically relevant unexpected findings or for those with benign unexpected findings (p=0.70 and p=0.77, respectively).

Comment #4

Please correct the statement or sequence of presented age (Patients who survived (201 patients, 70.1% of the overall population) were slightly but statistically significantly younger than those who did not survive (88 years vs. 86 years, respectively; p=0.036).

Reply #4

We thank the Reviewer for the comment. We have amended the text.

Modified text: page 4, line 160-161

Patients who survived (201 patients, 70.1% of the overall population) were slightly but statistically significantly younger than those who did not survive (86 years vs. 88 years, respectively; p=0.036).

Comment #5

Please change comas into periods in tables 

I hope this points would help improving the paper 

Reply #5

Thanks for the comment. We have changed the Tables accordingly.

Round 2

Reviewer 1 Report

Comments and Suggestions for Authors

The authors have addressed most of the reviewer’s recommendations, significantly enhancing the quality of their paper. 

Reviewer 3 Report

Comments and Suggestions for Authors

thank you for clarifcation